# Gamification in Tourism: A Design Framework for the TRIPMENTOR Project

Elina Roinioti [1], Eleana Pandia [1], Markos Konstantakis [2,*] and Yannis Skarpelos [1]

[1] Department of Communication, Media and Culture, Panteion University for Social and Political Science, 17671 Athens, Greece; eleni.roinioti@gmail.com (E.R.); eleana.pandia@gmail.com (E.P.); gskarp@panteion.gr (Y.S.)

[2] Department of Cultural Technology and Communication, Aegean University, 81100 Mytilene, Greece

* Correspondence: mkonstadakis@panteion.gr or mkonstadakis@aegean.gr

**Abstract:** In this paper, we discuss the gamification strategies and methodologies used by TRIPMENTOR—a game-oriented cultural tourism application in the region of Attica. Its primary purpose is to provide visitors with rich media content via the web and mobile environments by redirecting travellers, highlighting points of interest, and providing information for tour operators. Gamification is a critical component of the project; it relates users to specific sites and activities, improves their visiting experiences, and encourages a constant interaction with the application through a playful experience. In TRIPMENTOR, gamification serves both as a tourism marketing strategy and as a tool for encouraging users to share their experiences while exploring Attica in a way designed to meet their personal needs, interests, and habits. This paper aims to describe and analyse the gamification mechanisms applied, following the Octalysis framework, and discuss the opportunities and challenges of gamification as a tourist marketing strategy.

**Keywords:** gamification; Octalysis model; TRIPMENTOR; design; mechanics; tourism

## 1. Introduction

Attica, the Greek region including the capital city of Athens, is an international destination attracting tourists who wish to explore its 4000-year-old history, its monuments and landmarks, its museums, the well-known Athens Riviera, etc. It is also a destination for city break tourism, as well as a brief stop before reaching Greek islands or other destinations. It welcomed almost 6000 tourists on a daily basis during 2019—an increase of 4.3% compared to 2018. Revenue from tourism exceeded EUR 2.6 billion [1]. It was awarded the title of the "World's Leading Seaside Metropolitan Destination 2021" by the World Travel Awards [2]—an organisation recognising and awarding excellence in tourism, hospitality and travel. Despite its importance as a destination, the strategic plan adopted in 2016 realised that Attica is diverse and, therefore, previous attempts to promote it as a single destination were doomed to fail [3]. Thus, a novel approach to promoting tourism in the region and connecting it to memorable experiences was deemed necessary and formed the basis of the TRIPMENTOR project.

TRIPMENTOR aimed to develop a bilingual (Greek and English) mobile tourist guide for the Attica region, based on a grounded tourist typology, proposing personalised routes, and incorporating gamified features, to provide a unique and memorable experience to the visitor [4]. There were three innovative elements of the project:

(a) The development of a grounded, theoretically and empirically driven tourist typology, taking advantage of data pertaining to actual visitors, considering needs and travel motivations, and supporting the design of personalised narratives;

(b) The development of infrastructure collecting relevant data, providing a rich environment for end-users;

(c) The design of a gamified system supporting, enhancing, and creating memorable experiences.

The project aimed to (a) understand and meet the possible needs of its users and gain some insight into their mindsets; (b) help them get a good feel for the local culture, while taking advantage of prejudices and stereotypes to relate modern and ancient Greek culture; and (c) encourage interaction and exchange, aiming to invite visitors to immerse themselves in present-day Attica, pinpointing the picturesque, recalling the classic, and facilitating the discovery of the modern allure of everyday life.

This paper presents an in-depth look at TRIPMENTOR's gamification design, the choices made, its workflow, and its expected results. The paper is structured as follows: Section 2 examines the principles of gamification design and the state of the art in cultural tourism gamification. Section 3 outlines the methodology used to develop the gamification design, while Section 4 opens a discussion on digital storytelling and how it was implemented in our project. Finally, Sections 5 and 6 provide a summary and offer some conclusions.

## 2. Theoretical Analysis

In this section, the background knowledge necessary for understanding gamification design and its application in TRIPMENTOR, is presented.

### 2.1. Gamification

Gamification is usually described as a system applying game design elements to a non-game context to create playful experiences and modify users' attitudes while encouraging engagement and participation. Gamification may also be perceived as a form of service packaging, where a core service is enhanced by a rule-based system providing feedback and interaction mechanisms to users to facilitate, support, or even change their overall behaviour [5]. Gamification might prove to be a powerful tool for establishing brands, engaging users, and influencing their behaviour by adopting game mechanics beyond the traditional gaming context [6]. Features such as leaderboards and social interaction within a gamified context are positively related to the emotional, cognitive, and social forms of engagement [7]. Emotional engagement pertains to feelings of positivity and enjoyment when engaging with a gamified medium [8]; cognitive engagement entails focusing on or engrossment with the gamified medium [9]; behavioural engagement occurs when a user consumes energy, effort, and time during an interaction with other people involving a gamified medium [10]. Moreover, gamification is a valuable technique for brand management since it may increase brand equity [7]. Gamified marketing activities have a positive effect on satisfaction and brand loyalty, positive word-of-mouth, and resilience to negative publicity [11]. When applied in non-game contexts, gamification aims to provide incentive, motivation, excitement, agency, and purpose [12].

A distinction has been made between reward-based and meaningful gamification [13]. The former is a system designed to reinforce desirable behaviour by providing rewards (e.g., leaderboards, levels, and points). This is considered to be a form of extrinsic motivation and has a short-term effect on user behaviour. Conversely, meaningful gamification evokes intrinsic motivation, associated with transformative learning and long-term changes [14].

### 2.2. Gamification in Tourism

Tourism is about experiences, and tourism experience design aims to produce memorable experiences. Gamification as a part of such a design is inextricably connected to storytelling, i.e., in rendering the experience meaningful. Moreover, it leads to rewarding interactions, heightened joy, immersive experiences, brand recognition, and loyalty to the destination [15]. Therefore, tourists are receptive to a gamified experience during their visit [16]. Furthermore, gamification in tourism may be implemented in the physical world, with minimal use of technology, such as in most of the examples noted by Bulencea and Egger [17]. Technology, on the other hand, enhances tourism experience design with new affordances, taking advantage of geolocation, augmented reality, gamified travel tours for

urban and rural environments, theme park games, gamified immersive experiences in cultural heritage, gamification and transmedia storytelling, gamified restaurant experiences, gamification in hospitality, and gamified flight experiences [18].

Games in travel and tourism aim to educate the tourist and promote the destination, whilst location-based mobile games are designed to deepen visitors' engagement and immersion in the sociocultural environment. They are frequently modelled after the classic scavenger hunt, in which players must collect information while exploring for areas of interest, sometimes competing with other players [15].

Gamification is particularly recommended in zoos [19], museums [20], heritage sites [21], "green" and sustainable tourism [22], city tours [23], the hospitality and tourism industries [15,24–27], and the tourist sector as a whole.

Research on gamification in tourism is for the most part descriptive, showcasing success stories and best practices [28–30]. A few qualitative studies employ case studies [31,32] or focus groups [15], and very few employ research techniques.

Gamification may enhance a tourist service by transforming a city tour into a "slideshow" shareable with friends and family at home; increase customer involvement and the perceived value of the service, resulting in product and service differentiation; and provide enormous marketing possibilities in the tourist business. Travel has traditionally been the first sector to adopt innovative initiatives, while tourism is increasingly reliant on collaborative service production. Social media, mobile phones, and gaming provide technological tools for enhancing such experiences [33].

## 3. The TRIPMENTOR Design Methodology

Relying upon existing literature and research in the field of providing digital services to visitors, TRIPMENTOR was developed as an electronic guide to the Attica region. Its aim was to propose diversified and personalised experiences to both nationals and international visitors, opening opportunities for visiting lesser-known but still important points of interest. Gamification was implemented in the design of the application from the very beginning, as a means to promote positive user behaviours and enhance the overall visiting experience. In this section, we present our methodology in selecting a gamification framework, designing techniques, progress and feedback mechanics, the point and levelling system, and the scoring and social mechanics.

### 3.1. The Gamification Framework

The first step was the selection of an appropriate gamification framework that would guide our design choices. Over the years, different frameworks, based on technology-orientated or system-orientated approaches and user-centred designs, have been developed to meet different needs [34]. The MDA (standing for mechanics, dynamics, and aesthetics) framework [35], for example, is one of the best-known tools in game design and has been extensively used in gamification. Following an agile approach, the MDA offers an analysis of the game design process under three categories: rules, systemic processes, aesthetics, or emotional response. The framework is based on systemic logic rather than a user-centred perspective. Other models, such as the 6D [36] or the GAME framework [37], describe the steps for designing a gamified experience and the key information needed, based on players'/users' typologies [38].

We adopted the Octalysis model [39] for TRIPMENTOR, which is an open, dynamic framework for meaningful gamified experiences that aims to turn core users' drives into dynamic mechanics. This model may be applied both as a design tool and an evaluation tool for gamified applications, based on an eight-axis logic: eight parameters forming an octagon that a designer should consider during a gamification project. This octagon works as a point-based system where different design elements are represented by its eight edges: the upper part of the octagon includes the positive and motivational-driven features, or the so-called white-hat gamification, while the bottom part represents the negative and impulsive-oriented features (black-hat gamification). Each drive is assigned with points on

a scale from 0–10, where 0 means the total absence of the specific drive, while 10 signifies the perfect application. Note that the maximum score a project can receive is 800 points ($8 \times 10^2$) but, as Yu-Kai Chou mentions, how the points are divided in the respective categories is much more important than the final score [39].

The scoring system in Octalysis does not have the same meaning as in games, for example. Whereas in games everyone is trying to collect as many points as possible and every player is chasing the highest score [39], in the Octalysis model scoring is merely an indicator, and balance is the ultimate goal. As a result, even a design project scoring five points for each drive is fairly balanced and satisfies all basic drives. The Octalysis diagram allows the game developers to identify weak drives and proceed with the proper adjustments. For example, when the diagram is overly extended in the bottom core drives, an evaluation should be carried out to determine whether the design is mostly based on black-hat gamification. Including black-hat aspects, the Octalysis model motivates the designers to take them into account, instead of covertly ignoring them. Chou suggested that scoring is not set in stone but is rather a personal estimation based on subjective experience [39].

The point-based aspect of this model enhances the iterative procedure of design; after all of the design choices have been made, the designer can retrospectively evaluate the mechanics using the octagon and proceed with further adjustments. Developing a balanced design framework is the holy grail of gamification, and the Octalysis model has proven to be a valuable tool in the hands of designers.

Specifically, the eight axes or pillars can be briefly described as follows:

1. Epic Meaning: An essential motivational drive, making the players/users feel like they are part of an ecosystem, or belonging to the inner circle of a secret society. This motivation is usually built through an epic narrative framing users' activities and choices and making them part of a mission to achieve the greatest good.
2. Accomplishment: Perhaps one of the most recognisable and overused motivational drives. People need to set specific long- or short-term goals to accomplish a mission or a task and receive pleasure from it. Accomplishment refers to designing an experience in which the players/users know what the target is and how to achieve it and receive instant feedback on their choices. Badges, leaderboards, status points, and levelling systems are mechanics that guide players towards success.
3. Empowerment of Creativity and Feedback: Empowerment is highly connected to the aforementioned drives. The feeling of accomplishment cannot be cultivated unless the players/users are empowered. However, empowerment is also connected to creativity, and to the ability to indulge oneself in a process providing the freedom to play, create one's own content, express one's creativity, and evidence the results of one's actions. The instant feedback, boosters, and milestones are among the most usual game mechanics for feedback.
4. Ownership: For most users, the sense of creating something of their own, redeemable as a sort of digital currency, is essential. It cultivates the feeling that their efforts and, probably, their resources are well spent, boosting their loyalty to a specific service or brand.
5. Social influence: Sociality is one of the core motivational drives that every designer must consider. How to share information, how and in what ways users will interact, and how players can display their social influence are essential to design choices to satisfy this basic need for connectivity and sociability. The next three motivational drives of the Octalysis model differentiate black-hat from white-hat gamification. Black-hat gamification promotes short-term addictive behaviours, rather than long-term engagement.
6. Scarcity: Scarcity is related to exclusiveness, with an opportunity, a prize, or an event that is not globally accessible, due either to time limitations or to random factors that a user cannot control. In everyday life, we see people becoming anxious to get their hands on products available only for a limited time or to a specific audience.

7. Unpredictability: Unpredictability is similar to scarcity but highlights the effect of surprise when something pleasant yet random happens. In games, for example, rare items satisfy this drive. Nevertheless, if these rare items are found during looting and under commonly accepted conditions, such as after a successful battle, then we can conclude that this mechanic has a positive effect (white hat). On the other hand, if rare items are linked with loot boxes, promoting a more gambling-like gaming behaviour, it can be labelled as a black-hat mechanic.

8. Avoidance: Avoidance expresses the fear of not seizing an opportunity. As Yu-Kai Chou mentions, people identify a temporary opportunity and feel that they need to act quickly or else they will lose it forever [39].

### 3.2. Implementing Octalysis to TRIPMENTOR

The TRIPMENTOR project aimed to address three issues:

- Increasing user satisfaction. This aim was two-fold: on the one hand, it meant satisfaction with the app itself and its service in creating a memorable experience, while also entailing satisfaction with the actual experience as conforming to the user's expectations from their travel.
- Promoting long-term and repeated use of the application, especially when visiting the Attica region. Long-term use means that users should be prompted to continue using the app after their travel is over, either to remember their experience, or to share their memories via social media, while repeated use means that they are prone to re-use the app in their subsequent visits to the area.
- Enhancing user engagement (commitment) through social interaction.

To address these issues, we adopted the Octalysis model as a design framework. As shown in Figure 1:

1. Epic Meaning: TRIPMENTOR is based on the personalisation of the tourist experience, based on tourist typologies. Storytelling plays an essential role in guiding and framing TRIPMENTOR's design. However, specific constraints made the development of a narrative incorporating the features of "Epic Meaning" challenging. The narrative aims to support the point system and build the levelling system while controlling the interaction between users and the application. Thus, the "Epic Meaning" in TRIPMENTOR is limited.

2. Accomplishment: A specific point system and a narrative levelling system were used to promote achievement. Different indicators were employed to satisfy the need for progress and fulfilment.

3. Empowerment of Creativity and Feedback: To increase the motivation of creativity, TRIPMENTOR's design incorporates features such as "Create your own routes" and "Create your own games" for its users. These features are open to user-generated content; users may add their own content, create their own paths and minigames, and share them with the rest of the TRIPMENTOR community. Another feature included under the rubric of empowerment is map annotation, providing users with the resources to take notes about the places they have visited.

4. Ownership: Ownership in the digital world is a complicated issue since it is related to recognition rather than to possession per se. For TRIPMENTOR, design is linked to empowerment of creativity and feedback, as it allows the user to design/create a task/challenge and present themselves as the designer/creator. When users share a path, their name is displayed, and other users can vote for it as their favourite (heart point system). This vote is essentially a digital confirmation of the community praising one's digital "work".

5. Social Influence: TRIPMENTOR offers several opportunities to develop sociability among users, with functions such as "Post a secret", "Share a point of interest", or "Share a complete route on social media". As we describe in the relevant section, through these features, visitors can indicate alternative routes, propose new experiences, collect positive votes and, thus, gain the trust and acceptance of other users.

The social influence here is measured through a point-based system that calculates upvotes from other visitors.

6. Scarcity, Unpredictability, and Avoidance: Building motivation through the afore-mentioned drives essentially means using the devices of surprise and the unexpected in a positive way, building towards white-hat gamification. We used a function called "Secrets" as the element of surprise, and the "Mini games challenge" as an event-based limited-period initiative motivating participation. Secrets are geolocated algorithm-defined messages that pop up at specific points of interest. Users may also "organise" short-term mini gaming challenges, using ready-made forms provided by the application. In TRIPMENTOR, scarcity and unpredictability are satisfied through the "Secrets" feature, while avoidance is satisfied through the "Mini-games".

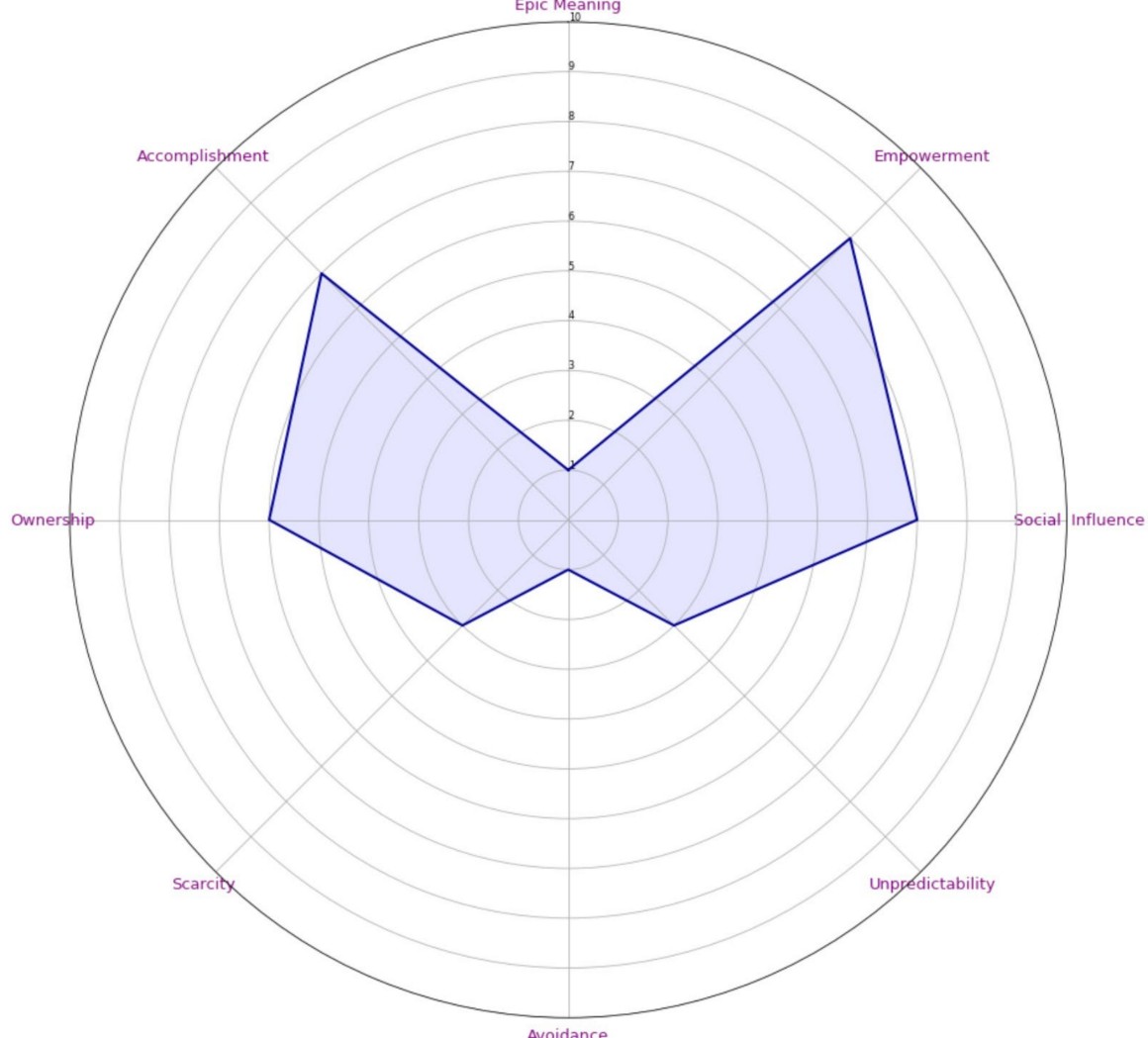

**Figure 1.** TRIPMENTOR Octalysis analysis.

A summary of gamification design characteristics in terms of the motivations they satisfy is provided in Table 1.

**Table 1.** Characteristics of TRIPMENTOR's gamification design.

| Characteristics | Motivation They Satisfy |
|---|---|
| Share a secret | Scarcity, Unpredictability |
| Share a secret (user-generated) | Creativity, Sociality |
| Create a route (user-generated) | Creativity, Ownership, Sociality |
| Social media sharing | Sociality |
| Levelling your avatar | Progress, Feedback, Accomplishment, Storytelling |
| Point system | Progress, Feedback, Accomplishment |
| Route completion progress bar | Feedback |
| Heart levelling system | Feedback, Sociality, Accomplishment |
| Create your own mini game | Creativity, Ownership, Sociality |
| User map annotation | Creativity |
| Minigame challenge | Sociality, Scarcity, Avoidance |
| Sociality index | Sociality, Feedback, Progress |

Using the Octalysis model as an evaluation tool, we were able to visualise our design choices. Figure 1 makes clear that our design depends heavily on white-hat gamification (we can observe the sides of the octagon extending to the right and left), while the bottom side is less developed. Our design scored 228 points, mainly distributed in the following drives: Accomplishment, Empowerment, Ownership, and Social Influence. The lack of the Epic Meaning motivation creates a schematic imbalance, highlighting the need to further explore the potential of building our "grand narrative". Scarcity, Unpredictability, and Avoidance, despite scoring low, are linked with white-hat gamification and this is an important aspect of our design. As evidenced in Table 1, they are connected to user-generated challenges, minigames, and features offering an engaging experience. In Figure 1, we can see how our gamification design (in blue) matches a perfect octagon. Analytically, Epic Meaning scored 1/10 points since we could not bind our gamification design with an epic narrative. Accomplishments scored 7/10, including mechanics such as "Level up your avatar", and different scoring systems, such as the heart levelling system. Empowerment of Creativity and Feedback received 8/10—the highest score of any category. TRIPMENTOR is heavily based on user-generated content ("Create a route", "User map annotation", "Share a secret", etc.), and that design choice is reflected in the individual scores. Ownership and Possession scored 6/10. Visitors spend time and energy creating their own content. This kind of effort generates interest in the result and, finally, satisfies the ownership drive. Social Influence is the second-highest scoring category, with 7/10 points, and mechanics such as the "Social index" and the "Mini-game challenge" create a measurable social experience. The bottom core drives, including Scarcity and Impatience, are less developed in our design. Scarcity and Curiosity scored 3/10 each because they can both be satisfied only through the "Share a secret mechanic". Finally, Loss and Avoidance received 1/10 points solely due to the "Mini-games challenge", whose short duration and quick start encourage participation.

In gamification design, different models and description schemes are used to depict, organise, and analyse the system flow and the end-user experience, emphasising different aspects of design. Specifically, in the recent literature, gamification patterns are used as design solutions to common design problems—from Herger's 10-pattern model [40] and the gamification pattern analysis by Ašeriškis and Damaševičius [41], to other more analytical formats, the interest of the research and design community is in how to systemise our common knowledge gained from games, and how to best apply it to projects where, as Swacha and Muszyńska (2016) [42] put it, "designers have the least freedom"—a freedom usually limited by business models, corporate goals, target groups, etc. In TRIPMENTOR, the Octalysis model was used as a gamification design model, but also as a concept documentation model. Its drive-based structure provides us with the design freedom to work with different audiences and types of visitors and address different kinds of problems.

In the following sections, we outline what kinds of mechanics we used, how they were implemented, what kind of aesthetic experience we expect to obtain, how they relate with other mechanics and, finally, what kinds of needs they were designed to meet. The TRIPMENTOR application is currently under development, so no public evaluation data can be disclosed at this point.

### 3.3. Mechanics

In general, mechanics relate to game rules, but mainly refer to the processes that support and frame a flow of actions, while at the same time providing the necessary feedback for the users to be able to enjoy and enhance the experience offered.

The mechanisms applied in TRIPMENTOR may be classified into the following categories:

(a) Progress and feedback mechanics: Mechanics that provide a sense of evolution and progress.
(b) Social mechanics: Mechanics that support user interaction and create an online community.
(c) Engagement mechanics: Mechanics that depict users' online actions and behaviours concerning the affordances provided.

### 3.3.1. Progress and Feedback Mechanics: Point and Levelling System

The levelling system is a system of rules regulating how the users can satisfy their needs for progress and development. In practice, the levelling system provides users with the tools to evolve to ever-higher levels of experience, collecting points and receiving perks. As expected, designing the levelling system also affects the end-user experience.

If the designer chooses to use a linear progression system, the player must make an extra effort to achieve the next level. For example, if a gamification system is based on collecting rubies to level up, and levelling from the first to the second level is completed when two rubies are collected, the next level will need four rubies, the fourth will need six, and so on.

An exponential growth system, on the other hand, gives players a quick start, but levelling becomes harder as they grow to the next level. This design choice helps players build their experience in the beginning when they require it most, but afterwards, it becomes increasingly demanding (Figure 2). Most RPG games are based on this logic.

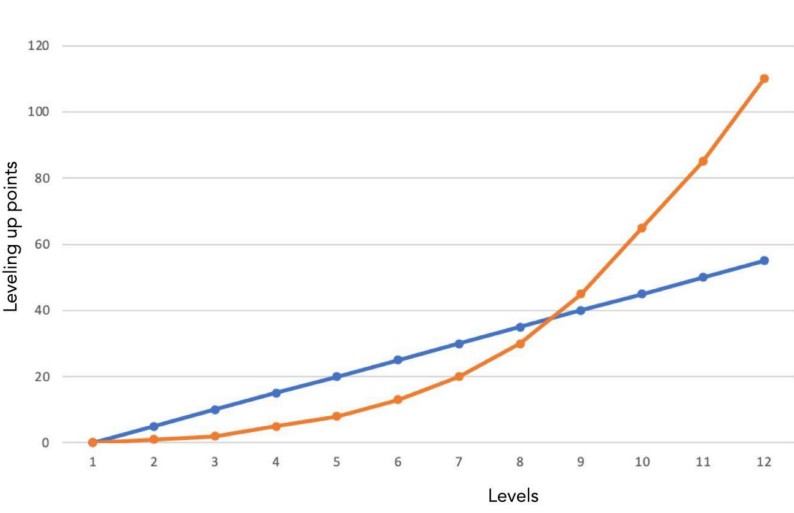

**Figure 2.** Exponential progression diagram (x = levels and y = levelling up points).

For TRIPMENTOR, we designed a levelling system based on the exponential progression to boost first-time users and satisfy their drive for achievement and progress. Following the exponential growth, the user has a quick and satisfying start, but the most creative and community-driven mechanics of the application may only be unlocked at later

levels. As users learn how to use the application and to what ends, they become even more motivated to use TRIPMENTOR, and level up, to gain access to mechanics that are available only to loyal members of the TRIPMENTOR community.

For the levelling system, we used the following formula:

$$E(v) = BaseXP * (b^v - 1)$$

where:

$E(v)$ = the experience a user needs to gain in order to progress from level $v$ to level $v + 1$;

$BaseXP$ = the linear difficulty coefficient;

$b$ = the exponential difficulty coefficient.

For example, if we define $b = 3$ and $BaseXP = 5$, then our levelling system will have the following form:

$$E(2) = 5 * \left(3^2 - 1\right) = 40 \quad E(3) = 5 * \left(3^3 - 1\right) = 130 \quad E(4) = 5 * \left(3^4 - 1\right) = 400.$$

Balancing is an important factor for games and gamified applications. Balance essentially means to present a kind of experience in which every aspect of the game serves a specific purpose in the best possible way. When it comes to games, this means, for example, that different weapons do different kinds of damage, promoting a sense of fairness to the game or the experience of flow to the user—the game is neither too difficult nor too easy for the user to handle. Balancing is hard to achieve, and it goes beyond the implementation of a mathematical formula, being the result of an iterative process of designing, testing, and refining.

In our case, balancing our system was mainly related to the exponential coefficient that would ultimately affect the effort a user has to make to increase levels. To decide our final formula, we had to run multiple scenarios and decide which was the best for our needs.

TRIPMENTOR's levelling system includes two easy-to-reach levels for the average 2-day visitor, one medium difficulty level for those who either have planned a more extended stay in Athens or are short-term visitors but want to explore TRIPMENTOR's gamified experience further, and finally, two high levels for the returning visitors and/or loyal users.

### 3.3.2. Scoring

Scoring became a crucial part of our design, specifically because we had to address the issue of defining and visualising a "Complete route". TRIPMENTOR is based on the idea of personalised routes that reflect and satisfy a visitor's age and gender, habits, and personality. Based on a user's profile, the guide will propose several routes so that they can enjoy a personalised tourist experience. Modelling how a user completes a route and, thus, a gamified task, affects the entire design.

When subscribing to TRIPMENTOR, users are invited to play a quick game that works like a fun questionnaire. Playing a fun minigame and choosing different activities and personal preferences, users create their profile, which will be a mixture of different tourist ideal types.

For instance, Maria is 35 years old and, according to her profile, she has a 50% interest in getting to know the city, the people, local culture, and local food; 40% of her interest is in archaeological sites and history, and 10% in exploring new areas and satisfying a sense of adventure. Based on our design, the application will propose several routes relevant to her primary profile (50%), another set of routes corresponding to her secondary profile (40%) and, finally, a group of routes for the last 10%. The rationale behind TRIPMENTOR's design is to motivate users to go one step further from their comfort zone, so as to experience the city through different lenses corresponding to different aspects of their personality. To achieve this, we rendered different values in different routes following a reverse logic from that of the users' personality profiles. Taking into account that it is very likely that users with or without the help of any application will visit points of interest relevant to their primary profile, TRIPMENTOR provides bigger rewards for choosing a route from

the secondary or even the tertiary profiles. In other words, the routes corresponding to the secondary profile receive a greater value and more points than those from the primary profile, etc. In this light, it is obvious by now that the same route will not have the same value for all profiles.

For Maria, who wants to indulge in Athenian everyday life, visiting the central market will give her a few points. On the other hand, for her friend John, who has a different and more athletic profile and wishes to visit sunny Athenian beaches, the same route will offer a higher value—a reward for trying something that matches his profile but not his first choices.

Of course, scoring is only part of the solution. We had to decide what constitutes a complete route and how we could provide extra motivation for users to interact with the app. Adding a fulfilment indicator as a visual reference for users' progress, along with a completion bonus, makes the experience even more engaging.

Generally, our scoring follows the following logic:

- If the users manage to visit 80% of the proposed routes, then they will receive x points;
- If the users visit only a few sites—less than 80%—then their scoring will be based on a specific formula that calculates the value of each point of interest and multiplies it by the total number of points of interest the user visited;
- If users visit all the sites of a specific route (100% completion), then they receive all the points, plus a bonus point.

### 3.3.3. Social Mechanics

Social mechanics aim to satisfy the user's drive for social contact, networking, and communication. They help build a sustainable community of people who share common interests. The electronic world of mouth (eWOM), interpersonal online communications, and suggestions originating from online friends, relatives, and distant partners have proven to be the primary resource for travel recommendations today. In fact, travellers use social media as their pool of information for recommendations and decision making not only before they embark on their journey, but also after their trip, for evaluating sites and services and sharing their personal experiences [43]. Providing proper mechanics to further foster this kind of information and experience sharing thus became an integral part of our design.

Our primary interest was in user-generated content, i.e., content created, produced, and distributed by the users. As has already been mentioned, studies have highlighted that when referring to user-generated content, it is crucial to keep in mind that the end-users should be motivated to produce it voluntarily by themselves, without any kind of reward [44]. On the other hand, gamification design is mainly based on behavioural changes and a kind of gaming enjoyment that derives from satisfying basic psychological needs such as relatedness [45]. Taking all the above into consideration, our goal was to give users the freedom/motivation to decide whether they want to make use of our user-generated features, and not to indirectly manipulate them into doing so through black-hat gamification techniques. In other words, our goal was to design a gamified system around those features, supporting users' participation in meaningful ways, and not to create a system taking advantage of their needs and expectations.

The social mechanics designed for TRIPMENTOR may be divided into low-level engagement mechanics, such as sharing information or a secret that one has discovered about a place or a site, and high-level engagement mechanics, such as creating one's own minigame. The effort and time required by a user to complete a social goal are the two factors that, ultimately, determine the points that each social goal will accumulate and, consequently, the level at which this social goal will be unlocked for users. Creating one's own minigame, for example, is far more demanding than posting a hint about a place that one recently visited and is also much more community-related. Unlocking this high-level engagement feature at a more advanced level, to some extent, provides a safety net for the designer, ensuring that this feature will not be misused by users who have no actual interest in the community and the application itself.

3.3.4. Engagement Mechanics

Engagement mechanics differ from social mechanics in being a quantifiable index of a user's interaction with the application and its community. In other words, while social mechanics are used to motivate people and satisfy specific needs such as sociality, creativity, and ownership, engagement mechanics operate as a public visualisation and reward for in-app presence, just like badges do. Studies on the use of badges as a motivational index have been extensive, highlighting positive effects such as behavioural changes with respect to specific practises and habits, social recognition, and social influence, as well as adverse effects, such as promoting behaviours only to collect more badges. Instead of applying a badge system in TRIPMENTOR, we decided to use two indicators interconnected with social mechanics depicting the user's relationship with the community.

The first indicator is a sociality index, mainly connected with the innovative features of the application design and user-generated content. Using a heart as a symbol, the sociality index positively represents the acceptance one enjoys from TRIPMENTOR's community of users. These hearts, which function as another levelling system with different colours, display the likes that the user has collected with respect to a route or a minigame that they created. The second indicator refers to users' extroversion or, in other words, users' engagement with social media, e.g., sharing posts or reposting information about TRIPMENTOR events.

## 4. Storytelling and Gamification in TRIPMENTOR

As is evident from the presentation so far, applying a closed narrative scheme was almost impossible in gamifying TRIPMENTOR. Attica's visitors (as with most visitors to major cities) come from different socioeconomic backgrounds and cultures, diverse age groups, and have diverse purposes for their visits. An archaeology student from abroad has little to nothing in common with a Greek middle-aged businesswoman who visits Athens regularly and has only a few free hours available to explore the city. On the other hand, the narrative provides an essential layer of experience that users need to shape their own experiences and create their own narratives. In TRIPMENTOR, we used storytelling (a) to further customise users' profiles and offer a personalised experience, and (b) to support the feeling of achievement and progress [46].

We used storytelling elements to develop different tourist personas for users to adopt and different environments for them to explore, based on their preferences. The colour palette, the general aesthetics, the wording, the text, and the interface design were the objects of extensive elaborative research that ran for a period of a year, aiming to create a personalised experience through storytelling. This part of TRIPMENTOR's research is beyond the scope of the present paper.

The second aspect of storytelling is connected to in-game progress and, thus, to our levelling system. The goal here was to design a narrative path to guide users from the first level to the last. "Levelling your avatar" is a visual progress mechanic whereby levelling unlocks new virtual avatar accessories that correspond to the progress made. New avatar accessories mean different items that the user's avatar can wear, and props that are only available at certain levels. In TRIPMENTOR, instead of using badges as a visualised progress indicator, we decided to use an avatar upgrade system; for example, an avatar with a map indicates a second-level user who has a lot to learn and explore, while a pair of glasses indicates a high-level user who knows their way around Attica. These kinds of visual indicators help to build storytelling for the users and, at the same time, symbolically reflect the experience of the city that they have gradually gained. After all, who would not want to unlock the final level and virtually wear a hat and a backpack, and become the professor, who has mastered and unlocked all the city's secrets and is ready to share their wisdom with others?

In conclusion, digital storytelling is not only about writing stories and scripts, but also about spatial design, and creating a holistic, consistent experience using different elements.

During the TRIPMENTOR research project, storytelling was part of the game experience that we meant to offer, aiming to further support our game design choices.

## 5. Discussion

A gamification design is an iterative process, and further testing will be needed to evaluate the provided experience. How fast or slow visitors level up, and how this affects their motivation to reuse the application and, therefore, further discover its perks and city secrets, is an open question for research, and is our next development step. From a design standpoint, what is worth mentioning, and practically guided our design methodology throughout the entire project, is the question of how to intrinsically motivate people to be part of a travel community using microgames, or in other words, small and quick gaming events/games.

Our entire design is based on this exact concept: developing mechanics and features that resemble micro gaming. The choose-your-type playful questionnaire, the thrill of the minigame challenges, and the unexpected share-a-secret feature were derived from gaming culture, aiming to enhance this non-gaming environment that we often discuss in gamification design. Therefore, our goal was not just to adopt gaming mechanics and apply them in a non-gaming environment such as a tourist application, but also to be inspired by gaming culture and to develop microgames that can engage, motivate, and entertain users.

For example, every digital application has an onboarding phase of interaction. Onboarding is the step during which the users have already discovered the application, expressed interest in trying it out, and are about to register and start using it, learn its rules, and test its interface. Yu-kai Chou refers to four steps of gamification experience, starting with (a) Discovery and (b) Onboarding, and continuing with (c) Scaffolding, in which users know exactly how to use the application and how to gain the maximum experience, and finally (d) the endgame, at which point users have mastered the application and need extra motivation to keep using it [39]. In TRIPMENTOR, the onboarding step is the playful questionnaire—a feature that we believe is casual enough to make new users feel at ease and, at the same time, give a fun aspect to a non-fun activity (registering and creating users' profiles). Through gaming elements and micro gaming, onboarding and all the other gamification experience steps can be achieved effortlessly, hopefully helping designers to build a new communicating path with their users/players.

## 6. Conclusions

Creating a gamification scheme for a tourist destination is quite complex since several actors are active without necessarily promoting a unitary narrative or brand—a unique experience for the visitor agreed upon by all of them. Promoting a region as a destination is even more complicated, since actors and agents at different hierarchical levels interact, and opposing interests are at play, while imbalances in development can deepen the imbalances in the number and appeal of points of interest.

Successful gamification in tourism does not necessarily imply a digital application. However, a mobile application may highly enhance its success in augmenting interactivity. For example, geolocated information augments the cognitive and emotional responses to the point of interest; routes and gamification provide a narrative thread connecting discrete places and allowing for feeling a purpose in the visit—even in a purposeless stroll in the city or the countryside; reading other users' comments, proposals, or secrets creates a sense of community—a kind of intimate knowledge shared between co-travellers; personalisation through tourist typology brings the user to the centre of the tourist experience, catering to their priorities while at the same time challenging them and opening up a different perspective when proposing routes or secrets for their secondary or tertiary type, and rewarding the user for choosing them; the hearts as a sign of recognition coming from the community, and not from the application alone, is a pride-fuelling choice. However, all of these aspects look like scattered pieces of a puzzle in need of a uniting principle—a systemic logic to pull the threads.

It was this principle—the framework that we described—which first led to the design of mechanics interconnecting actions, information flow, and experiences.

In this paper, we detail the design of the rule-based systems and the gamification framework for the TRIPMENTOR project, as a methodology that may be followed in gamification design, irrespective of the actual content. This is why we did not confine ourselves to presenting the choices made but insisted on presenting the choices available. Having an idea of the whole spectrum of possibilities, a gamification designer may proceed to different choices relevant to the specifics of their project, the necessities of their content, and the expectations for its outcomes. We also followed the design logic from the framework for the rules, to calibration, thus presenting the concentric layers in building a gamification project.

The choice of the Octalysis model as our framework allows for better understanding of to what extent our gamification covers crucial aspects of the project, as well as for evaluating design gaps that may be answered in revisions, when enough data about real users' behaviour will be available. It also allows for the determination of to what extent critical choices are implemented (e.g., using white-hat and avoiding black-hat gamification).

We then turned to gamification mechanics, aimed at providing a sense of evolution and progress, supporting user interaction, and creating an online community, as well as mildly rewarding users' actions and behaviour. Our concern was to provide interactive and fun features as a solution to a specific travelling "problem": what is the best route to follow based on one's personality traits, the purpose of the trip, and one's visiting habits? User-generated content management mechanics were employed to support the sense of community, to promote interactivity among users, and to provide extra tools for the users to express themselves and share their experiences. User-generated content is also crucial for content updates. By providing the users with the tools to create content, the application satisfies a common need for sharing knowledge, while building a pool of constantly updated material. Nevertheless, user-generated content comes with limitations. Users may post illegal content, behave in a non-acceptable manner, harass people, or even use the application in a way that is harmful to the brand and the community. In the case of TRIPMENTOR, these problems are addressed partly through the levelling system, since access to user-generated content features is limited to competent users, and through the provision for a content moderator.

As it stands TRIPMENTOR is about to begin its social career among visitors. The data collected after an initial period of real-life testing will allow for an evaluation and calibration of the gamification design, leading to the evaluation of the theoretical background based on the foundation of hard experience.

**Author Contributions:** Conceptualisation, E.R. and M.K.; methodology, E.R.; software, E.R. and M.K.; validation, Y.S.; formal analysis, E.P.; investigation, E.R. and E.P.; resources, E.R.; data curation, E.R.; writing—original draft preparation, E.R., E.P. and M.K.; writing—review and editing, E.R., E.P. and M.K.; visualisation, Y.S.; supervision, Y.S.; project administration, M.K.; funding acquisition, M.K. All authors have read and agreed to the published version of the manuscript.

**Funding:** This research was co-financed by the European Union and Greek national funds through the Operational Program Competitiveness, Entrepreneurship, and Innovation, under the call RESEARCH CREATE INNOVATE (project code: T1EDK—03874).

**Institutional Review Board Statement:** Not applicable.

**Informed Consent Statement:** Not applicable.

**Data Availability Statement:** Not applicable.

**Conflicts of Interest:** The authors declare no conflict of interest.

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
