# Peer review of "Gamification in Tourism: A Design Framework for the TRIPMENTOR Project"

_digital, doi:10.3390/digital2020012_

Round 1

Reviewer 1 Report

The reviewed work was defined as a scientific article, but it does not have the proper structure. The methodological chapter is followed by conclusions. There is no chapter on results and discussion. I propose to separate these issues from chapter 3.
There are two different ways to refer to the publications cited. The scope of the cited literature is sufficient.
The discussed topic seems to be interesting and absorbing, as such an approach is rarely found in the literature. I propose to introduce the suggested changes, which may improve the scientific value of the work.

Author Response

Dear Editor,

Thank you for the opportunity to revise our manuscript “Gamification in Tourism: A Design Framework for the TRIPMENTOR Project”. We appreciate the careful review and constructive suggestions. It is our belief that the manuscript is substantially improved after making the suggested edits, highlighted within the document by using coloured text.

Reviewer #1 

R1#1: The reviewed work was defined as a scientific article, but it does not have the proper structure. The methodological chapter is followed by conclusions. There is no chapter on results and discussion. I propose to separate these issues from chapter 3.

The aforementioned changes have been addressed in the revised manuscript. Additionally, we modify the type of document to a concept paper.

R1#2: There are two different ways to refer to the publications cited. The scope of the cited literature is sufficient. 

We prepare the references with a bibliography software package (ReferenceManager) to avoid typing mistakes and duplicated references. We include the digital object identifier (DOI) for all references where available.

R1#3: The discussed topic seems to be interesting and absorbing, as such an approach is rarely found in literature. I propose to introduce the suggested changes, which may improve the scientific value of the work.

The aforementioned changes have been addressed in the revised manuscript. 

We thank R1 for his response.

Reviewer 2 Report

I have read with great initial interest the manuscript entitled 'Gamification in Tourism: A Design Framework for the TRIP-2 MENTOR Project'. The topic of new information technologies applied to tourism is of great interest and opens up many research possibilities.

It is an article with potential for publication. However, I think it needs to be strengthened in the following aspects:

The text is not really a research, it simply explains a project (TRIP-2 MENTOR Project). It is a work of synthesis.

Summary only there are objectives, it should also include method, most important results and conclusion.

The paper is poorly structured. There are several sections that must be repositioned because the logical order of an article is not followed. I recommend the following order: 1. Introduction; 2. Theoretical analysis; 3. Materials and methods, 3.1. Study area…; 4. Results and discussion; 5. Conclusions; 6. References.

A map of the area under analysis should be included in the study area section, which is not included in this work.

The conclusions are very simple and summarize what was previously described.

Author Response

Dear Editor,

Thank you for the opportunity to revise our manuscript “Gamification in Tourism: A Design Framework for the TRIPMENTOR Project”. We appreciate the careful review and constructive suggestions. It is our belief that the manuscript is substantially improved after making the suggested edits, highlighted within the document by using coloured text.

Reviewer #2 

R2#1: The text is not really research, it simply explains a project (TRIPMENTOR Project). It is a work of synthesis.

The aforementioned changes have been addressed in the revised manuscript and we emphasise the research aspects of the project. Additionally, we modify the type of document to a concept paper.

R2#2: Summary only there are objectives, it should also include method, most important results and conclusion.

The aforementioned changes have been addressed in the revised manuscript.

R2#3: The paper is poorly structured. There are several sections that must be repositioned because the logical order of an article is not followed. I recommend the following order: 1. Introduction; 2. Theoretical analysis; 3. Materials and methods, 3.1. Study area…; 4. Results and discussion; 5. Conclusions; 6. References.

The aforementioned changes have been addressed in the revised manuscript.

R2#4: A map of the area under analysis should be included in the study area section, which is not included in this work.

The aforementioned changes have been addressed in the revised manuscript.

R2#5: The conclusions are very simple and summarise what was previously described.

Three paragraphs have been added in Section Conclusions. 

We thank R2 for his response.

Reviewer 3 Report

A well-written article that explains well, in general and the concept of the authors' own application, the advantages and method of implementation of gamification in tourism. The article needs to be edited formally.

Formal comments:

  • do not use footnotes, can be replaced by parentheses:

1 Official link: https://www.worldtravelawards.com/

  • do not use footnotes, can be replaced by parentheses and by reference to the described source:

2 “Draft plan for the development of tourism 2021-2025”, Hellenic Ministry of Tourism, Available here:
https://mintour.gov.gr/wp-content/uploads/2021/05/tomeako-programma-anaptyksis-tourismou-2021-2025.pdf
3 A brief report of the program, scopes and partners can be found here Vassilakis, Costas, et al. "TripMentor Project:
Scope and Challenges." CI@ SMAP. 2019.

5 You can find an online version here: https://yukaichou.com/octalysis-tool/

  • instead of a footnote, give a reference to the described source

4 https://balticmuseums.info/

  • it is necessary to describe the axes of graphs - units and what is plotted on them

There are extra hard spaces in the text, e.g. here: "firms make intense use of   crowdsourcing, perceiving"

for direct citation, there should be a uniform format throughout the work and mark it with quotation marks and italics, i.e. also here:

Additionally, Sigala [13] notes that "despite their differences in emphasis, all definitions include both a systemic component that explains how the game is produced (e.g., the usage of game mechanics) and an experiential component that characterises human interaction and results inside the game.

used alternatively "e.g." and also "and others", not both at once in the same list of options

As such, TRIPMENTOR intends to employ gamification to promote positive user behaviours (e.g., motivation, involvement, and cooperation, and so on

here better "their concept", or here again give a link to the source

The basic tenet of Huotari and Hamari's concept is that

here too the link must be in the format [1]:

"Avoidance: Avoidance expresses the impulse of not seizing an opportunity. As Yu-Kai Chou mentions, people identify a temporary opportunity and feel that they need to act quickly or else, they will lose it forever (Yu-Kai, 2014)."

the use of [] is confusing and interchangeable with reference to the source

the narrative aims at [1] supporting the point system and [2] building the levelling system, while simultaneously 3] defining the general framework of the interaction between users and the application

better explain the acronym:

Most RPG games are based on this logic.

Author Response

Dear Editor,

Thank you for the opportunity to revise our manuscript “Gamification in Tourism: A Design Framework for the TRIPMENTOR Project”. We appreciate the careful review and constructive suggestions. It is our belief that the manuscript is substantially improved after making the suggested edits, highlighted within the document by using coloured text.

Reviewer #3 

R3#1: do not use footnotes, can be replaced by parentheses.

The aforementioned changes have been addressed in the revised manuscript.The footnotes changed to citations.

R3#2: it is necessary to describe the axes of graphs - units and what is plotted on them

The aforementioned changes have been addressed in the revised manuscript.

R3#3: There are extra hard spaces in the text and for direct citation, there should be a uniform format throughout the work and mark it with quotation marks and italics.

The aforementioned changes have been addressed in the revised manuscript.

R3#4: use alternatively "e.g." and also "and others", not both at once in the same list of options.

The aforementioned changes have been addressed in the revised manuscript.

R3#5: the use of [] is confusing and interchangeable with reference to the source

The aforementioned changes have been addressed in the revised manuscript.

We thank R3 for his response.

Round 2

Reviewer 2 Report

After reading the responses to my comments and reviewing the document again, I believe it is much improved and I accept its publication. It is appreciated that the authors have considered all my comments.

Author Response

Dear Reviewer,

thank you for the opportunity to revise our manuscript “Gamification in Tourism: A Design Framework for the TRIPMENTOR Project”. We appreciate the careful review and constructive suggestions.